# Metabolic Effects of Bovine Milk Oligosaccharides on Selected Commensals of the Infant Microbiome—Commensalism and Postbiotic Effects

**DOI:** 10.3390/metabo10040167

**Published:** 2020-04-24

**Authors:** Louise M. A. Jakobsen, Maria X. Maldonado-Gómez, Ulrik K. Sundekilde, Henrik J. Andersen, Dennis S. Nielsen, Hanne C. Bertram

**Affiliations:** 1Department of Food Science, Aarhus University, Agro Food Park 48, 8200 Aarhus N, Denmark; uksundekilde@food.au.dk (U.K.S.); hannec.bertram@food.au.dk (H.C.B.); 2Department of Food Science and Technology, University of California, One Shields Avenue, Davis, CA 95616, USA; mxmaldonado@ucdavis.edu; 3Arla Food Ingredients, Sønderhøj 10, 8260 Viby J, Denmark; hejan@arlafoods.com; 4Department of Food Science, University of Copenhagen, Rolighedsvej 26, 1958 Frederiksberg C, Denmark; dn@food.ku.dk

**Keywords:** gut microbiota, infant nutrition, microbial interaction, co-culture, metabolic activity

## Abstract

Oligosaccharides from human or bovine milk selectively stimulate growth or metabolism of bacteria associated with the lower gastrointestinal tract of infants. Results from complex infant-type co-cultures point toward a possible synergistic effect of combining bovine milk oligosaccharides (BMO) and lactose (LAC) on enhancing the metabolism of *Bifidobacterium longum* subsp. *longum* and inhibition of *Clostridium perfringens.* We examine the interaction between *B. longum* subsp. *longum* and the commensal *Parabacteroides distasonis*, by culturing them in mono- and co-culture with different carbohydrates available. To understand the interaction between BMO and lactose on *B*. *longum* subsp. *longum* and test the potential postbiotic effect on *C. perfringens* growth and/or metabolic activity, we inoculated *C. perfringens* into fresh media and compared the metabolic changes to *C. perfringens* in cell-free supernatant from *B. longum* subsp. *longum* fermented media. In co-culture, *B. longum* subsp. *longum* benefits from *P. distasonis* (commensalism), especially in a lactose-rich environment. Furthermore, *B. longum* subsp. *longum* fermentation of BMO + LAC impaired *C. perfringens*’ ability to utilize BMO as a carbon source (potential postbiotic effect).

## 1. Introduction

The infant immune system relies on human milk components for protection against infections [1]. Sialylated oligosaccharides from human milk protect against pathogens in the intestine of breastfed infants through synergistic mechanisms with the gut microbiota by stimulation of bifidobacteria and *Bacteroides* in the gut [1] and through prevention of pathogen adhesion to the intestinal surface [2]. Sialylated oligosaccharides are abundant in human colostrum and mature milk [2], but decrease with lactational age [3]. Infant formula contains only trace amounts of oligosaccharides [2], leading to microbiome differences between breast-fed and formula-fed infants. Bovine milk oligosaccharides (BMOs) are a rich source of sialylated oligosaccharides, especially 3′-sialyllactose (3′-SL), but also 6′-sialyllactose (6′-SL) [2]. The infant gut microbiome is dominated by bifidobacteria and lactobacilli, but also harbor commensals such as *Bacteroides*/*Parabacteroides* [4,5,6,7]. Several studies provide evidence of positive interactions between selected bifidobacteria and *Bacteroides/Parabacteroides* species [8,9,10] but it remains poorly understood how these interactions are linked to the specific source of fermentable carbohydrates. Elucidating the role of BMOs in modulating the infant microbiome composition might pave the way for using BMOs as a safe ingredient in infant formula. In a recent study [11], we found carbohydrate-dependent effects in complex co-cultures of the pathogenic *Clostridium perfringens* and *Parabacteroides distasonis* or *Bifidobacterium longum* subsp. *longum*. *P. distasonis* growth was positively associated with access to bovine milk oligosaccharides (BMO) and *B. longum* subsp. *longum* growth was positively associated to a combination of BMO and lactose (LAC). In addition, a combination of lactose and BMO negatively influenced *C. perfringens* growth in co-culture. Consequently, we speculate that there is a synergistic effect of combining BMO and lactose, which stimulates *B. longum* subsp. *longum* growth and simultaneously inhibits *C. perfringens* growth in a complex co-culture. In the present study, we aimed to investigate the potential interaction between *B. longum* subsp. *longum* and the commensal *P. distasonis* and the effect of carbohydrate availability by culturing them in mono- and co-culture. In addition, we aim to decipher the role of BMO and lactose in *B. longum* subsp. *longum* mediated inhibition of *C. perfringens* by inoculating *C. perfringens* into a cell-free supernatant of *B. longum* subsp. *longum* fermented lactose and BMO, i.e., elucidating any potential postbiotic effect on *C. perfringens* metabolism. Postbiotics effects include, among other microbe-mediated host-beneficial effects: the local, antimicrobial effects by e.g., short chain fatty acids and organic acids [12].

## 2. Results and Discussion

Undigested carbohydrates from human or bovine milk have a profound influence on the complex interplay between bacteria associated with the lower gastrointestinal tract of infants [11,13,14]. Here, we investigated a possible interaction between *B. longum* subsp. *longum* and the gut commensal *P. distasonis* when grown in co-culture. Finally, we examined the potential synergistic effect of lactose and BMO on *B. longum* subsp. *longum* growth and metabolic activity and the potential postbiotic inhibition of *C. perfringens* in cell-free media from *B. longum* subsp. *longum*. Annotated, representative ^1^H NMR spectra of the three bacteria included in the present study are shown in Figure 1, using the BMO + LAC treatment as an example.

### 2.1. A Potential Synergistic Effect of BMO and Lactose on Stimulation of B. longum subsp. longum Metabolism

Analysis of the purified BMO product showed that 3′-SL and 6′-SL were the main components (37 and 6% *w*/*w* respectively), while other oligosaccharides and lactose were only present in trace amounts (data not shown). In the BMO treatment, *B. longum* subsp. *longum* does not appear to utilize 3′-SL or 6′-SL or produce major metabolite quantities (Appendix A), but it does show growth compared to minimal media (Figure 2) after 24 and 48 h (except co-culture after 24 h).

Unlike *B. longum* subsp. *infantis* (ATCC 15697), *B. longum* subsp. *longum* (ATCC 15707) does not have the genes necessary for expression of a sialidase [15]. Based on previous research, the inability of utilizing 3′-SL and 6′-SL is a common trait in other infant strains of *B. longum* subsp. *longum* [16]. As expected, *B. longum* subsp. *longum* is able to ferment LAC as indicated by the conversion of lactose carbon into acetate and formate (Appendix A) and positive growth (Figure 2).

Interestingly, the *B. longum* subsp. *longum* metabolite production seems to be enhanced in the presence of both BMO and LAC. This observation is based on higher conversion of lactose into acetate, formate, and lactate (Figure 3), indicating a positive feedback mechanism on lactose utilization when in a BMO-rich environment. In addition to this, we speculate whether the higher metabolite production might be a result of *B. longum* subsp. *longum* metabolism of other less abundant oligosaccharides in the BMO product. *B. longum* subsp. *longum* isolates grew well on lacto-N-tetraose and lacto-N-neotetraose [16], neutral HMOs, that are found in minor amounts in bovine milk [17], but not quantified in the current study. According to the complete genome, *B. longum* subsp. *longum* ATCC 15707 (GenBank accession number AP010888.1) contains annotated genes for hydrolysis of neutral oligosaccharides with type I core structure (endo-α-N-acetylgalactosaminidase) and putative β-hexosaminidases. More work is needed to elucidate the impact of BMO components on lactose and oligosaccharide utilization in *B. longum* subsp. *longum* (ATCC 15707).

In a previous study, pure BMO appeared to stimulate *P. distasonis* growth in a complex infant-type co-culture [11]. Little is known about sialic acid utilization by *P. distasonis*, but a comparison of genomic sequences to *Bacteroides fragilis* revealed sequence similarities in the nanLET gene cluster and encoding of a nanH homolog (sialidase) [18]. The genome of *P. distasonis* ATCC 8503 (GenBank accession number AB238922) contains one candidate sialidase and one putative exo-α-sialidase as well as several candidate β-N-acetylhexosaminidases. Consequently, *P. distasonis* should be able to hydrolyze sialic acid from siallylactose and hexoses from neutral oligosaccharides and utilize them as carbon and energy sources when grown on BMO and BMO + LAC. *P. distasonis* cultures with BMO showed an average increase of 1.06 logs after 24, while 48 h growth was not above detection limit. However, due to the large difference in the maximum cell numbers reached in each replicate, overall growth did not reach significance (Figure 2). Nonetheless, *P. distasonis* releases minor amounts of lactose to the supernatant (Appendix A), which is an indication of utilization of lactose-bound carbon. The *P. distasonis* (ATCC 8503) genome contains candidate β-galactosidases, and indeed, in LAC fermentations, *P. distasonis* hydrolyzes lactose, which is evident in the glycan data, where it releases galactose and glucose to the supernatant (Appendix A).

### 2.2. P. distasonis and B. longum subsp. longum Co-Culture Increase Utilization of Lactose

Previous results from co-cultures of multiple organisms including *P. distasonis* and *B. longum* subsp. *longum* have suggested a substrate-dependent relationship between these two bacteria [11]. The current data suggests that commensalism occurs between *B. longum* subsp. *longum* and *P. distasonis*, as indicated by the changes in concentration of glycans in the supernatant of mono- and co-culture fermentation of LAC and BMO + LAC. Figure 3 presents the glycan and short-chain fatty acid composition as carbon mass. It is evident from carbon mass of glycan and short-chain fatty acids that growth of *B. longum* subsp. *longum* + *P. distasonis* co-culture during fermentation of LAC (Appendix A) and BMO + LAC media (Figure 3, Appendix A), result in a higher proportion of lactose carbon converted into acetate, formate, and lactate, indicating higher metabolic activity of *B. longum* subps. *longum* in co-culture (BL + PD) compared to mono-culture (BL or PD). This is especially evident in BMO + LAC treatment (Figure 3). While *P. distasonis* appears to hydrolyze lactose extracellularly as indicated by the release of galactose and glucose into the supernatant (Appendix A), *B. longum* subsp. *longum* most likely transports lactose into the cell and metabolizes it intracellularly. Interestingly, it appears that *B. longum* subsp. *longum* + *P. distasonis* co-cultures enhance fermentation of LAC and positively influences *B. longum* subsp. *longum* growth in 24 h co-culture samples (Figure 2). Even though *B. longum* subsp. *longum* is capable of utilizing galactose and glucose monomers released by *P. distasonis,* it is likely that they are not the preferred substrate when lactose is also available in the media. This is in accordance with data from another *B. longum* subsp. *longum* strain (NCC2705) that showed a preference for lactose rather than glucose as a substrate for carbon and energy metabolism, which could directly be traced back to lactose repression of the glucose-specific permease (glcP) [19]. The higher concentration of acetate and lactate in the co-culture with enhanced fermentation of LAC (Appendix A) is also an indication of enhanced *B. longum* subsp. *longum* metabolism. Fermentation of BMO + LAC by *B. longum* subsp. *longum* shows a similar metabolic pattern as in the fermentation of LAC, but the concentrations of acetate and lactate become even higher (Figure 3, Appendix A): this is especially pronounced in the co-cultures. Growth of *B. longum* subsp. *longum* or *P. distasonis* are not significantly different between mono- or co-culture fermentation of BMO + LAC (Figure 2). *B. longum* subsp. *longum* and *P. distasonis* mono- and co-cultures show very low metabolic activity when grown solely on BMO as indicated by low SCFA production and (very) limited change in glycans (Appendix A).

### 2.3. Through Metabolism of BMO + LAC B. longum subsp. longum Potentially Inhibits C. perfringens’ Ability to Metabolize Sialyllactose

*C. perfringens* was able to utilize BMO and LAC for growth. *C. perfringens* completely depleted 3′-SL and 6′-SL in the BMO media (Figure 4A,B), with a concomitant minor increase in lactose content (Figure 4C).

This finding indicates that *C. perfringens* utilizes the sialic acid monomer by a sialidase acting on the 2,3 and 2,6 glycosidic bonds of the sialyllactose units with subsequent lactose release. It is known that most bacterial pathogens carry genes encoding for sialidases to be able to harvest sialic acid from mucus [20]. Bacterial utilization of sialic acid from sialyllactose requires a sialidase, which is encoded by NanH, NanI, NanJ, or orthologous genes [21]. Previous studies found what appears to be a highly conserved sialidase encoding ORF (open reading frame) in many strains of *C. perfringens* [22,23] and, in fact, according to the sequence and annotated genome of *C. perfringens* ATCC 13124^T^ (GenBank accession number CP000246) does indeed encode three sialidases (nanH, nanI, and nanJ). *C. perfringens* fermentation of LAC resulted in a lower lactose concentration compared to uninoculated media (-,-) (Figure 4C) and minor increase in galactose (Figure 4D). The genome of *C. perfringens* ATCC 13124 (GenBank accession number CP000246) contains annotated β-galactosidase, sugar transporters, and sugar ABC transporters. Collectively our data and literature indicate a selective utilization of the glucose monomer from lactose, while uptake of lactose or galactose needs further investigations to decipher the catabolism pathways.

Growth (Figure 5) and butyrate production (Figure 6B) support the finding that *C. perfringens* is capable of utilizing BMO and lactose components as carbon sources. Depletion of 3′-SL and 6′-SL (Figure 4A,B) and a minor increase in galactose (Figure 4D) suggest that *C. perfringens* utilizes all the sialic acid and some lactose-derived glucose in the BMO + LAC treatment. Despite notable *C. perfringens* growth and metabolism in BMO and LAC treatments, the combination of BMO + LAC resulted in *C. perfringens* growth that was not significantly different from growth on minimal medium (Figure 5). However, the lower concentration of butyrate and acetate are indicative of a lower metabolic activity of *C. perfringens* in the BMO + LAC treatment (Figure 6A,B) compared to the pure BMO or LAC treatments. This cannot be attributed to acid accumulation generated during *C. perfringens* metabolism, as pH was comparable in *C. perfringens* fermentation of BMO + LAC (6.19 ± 0.23), BMO (6.06 ± 0.03), and LAC (6.37 ± 0.69). Most likely, the lower growth of *C. perfringens* is a result of a lower total carbon concentration in the BMO + LAC media, as the BMO component contained only 44% oligosaccharides.

The potential synergy between BMO and lactose on *B. longum* subsp. *longum* metabolism is of considerate interest. Not only does BMO + LAC increase metabolic activity of *B. longum* subsp. *longum*, but additionally the cell-free media from *B. longum* subsp. *longum* fermentation of BMO + LAC appeared to inhibit *C. perfringens*’ ability to utilize sialic acid from 6′-SL or 3′-SL (Figure 4A,B). The absence of butyrate in the resulting *C. perfringens* supernatant (Figure 6B) indicates a decrease in the metabolic activity of *C. perfringens*. This suggests a *B. longum* subsp. *longum*-metabolite or pH mediated inhibition of *C. perfringens*’ capacity to obtain sialic acid from 3′-SL and 6′SL, possibly through direct or indirect effects on the sialidase activity or expression. The pH of the cell-free media *B. longum* subsp. *longum* fermentation of BMO + LAC was low (5.42 ± 0.30) and even lower after *C. perfringens* fermentation (4.98 ± 0.14). The activity of *C. perfringens* sialidases are pH-sensitive, but the optimum is around pH 5–5.5 [24,25], and we therefore question if pH is the direct reason for inhibited sialidase activity. Sialidase production in *C. perfringens* is complicated, but it appears to be indirectly regulated by the VirS/VirR system, which is also an important regulator or virulence genes in *C. perfringens.* As an alternate hypothesis, we speculate that the low pH and organic acids in *B. longum* subsp. *longum* cell-free media, decreases sialidase expression through effects on the virulence gene expression. In *Clostridium difficile*, the virulence gene expression is pH-sensitive (lower expression at lower pH) [22,23,26]. If there is a similar pH-dependent effect on virulence gene expression in *C. perfringens*, a low pH might have downstream effects on sialidase production.

## 3. Materials and Methods

### 3.1. Strains and Propagation

This study included three bacterial strains: *Parabacteroides distasonis* (ATCC 8503), *Clostridium perfringens* (ATCC 13124), and *Bifidobacterium longum* subsp. *longum* (ATCC 15707). Frozen stocks were activated and propagated in rich media: *B. longum* supsp. *longum* in MRS (Thermo Scientific Oxoid, Waltham, MA, USA) with 0.05% L-cysteine and *C. perfringens* and *P. distasonis* in Brain Heart Infusion (Thermo Scientific Oxoid, Waltham, MA) in an anaerobic chamber operating at 37 °C with the following gas composition: 7% CO_2_/3% H_2_/90% N_2_. Growth curves under anaerobic conditions of each strain was determined at 37 °C by measuring optical density at 600 nm in an automated plate reader (BioTek, Winooski, VT, USA) for microtiter plates placed in an anaerobic chamber.

### 3.2. Lactose and BMO Treatments

This study included two carbohydrate sources: pure lactose and purified bovine milk oligosaccharides (BMO, Arla Food Ingredients Group P/S, Aarhus, Denmark). The BMO and minimal media composition was described in detail elsewhere [11]. Minimal media with 1% (*w*/*v*) lactose (termed LAC), BMO (termed BMO), or BMO and lactose (termed BMO + LAC) as the sole carbohydrate source were used for fermentation experiments. The media were pre-reduced by placing in an anaerobic chamber at room temperature overnight.

### 3.3. Standard Curves

Standard curves were obtained by inoculating a single colony into fresh rich media and incubated at 37 °C in a heating cabinet placed in an anaerobic chamber. Cells were harvest at the late exponential phase to maximize the CFU/mL and minimize dead cells. Standard curve cultures were diluted 10^−4^–10^−7^ in sterile saline (0.9% NaCl), plated onto agar plates and incubated anaerobically at 37 °C until counts were stable (48–72 h) and log CFU/mL calculated using the average count of colonies for the most populated plates with counts >20 and <250. The undiluted pure cultures were frozen at −80 °C for later DNA extraction and qPCR quantification.

### 3.4. Fermentation of Lactose and BMO in Mono- and Co-Culture

This study consists of three sub-experiments: (1) mono-cultures of *P. distasonis*, *B. longum* subsp. *longum*, and *C. perfringens* in LAC, BMO, or BMO + LAC (24 h), (2) co-culture of *P. distasonis* and *B. longum* sups. *longum* in 1% carbohydrate media (24 and 48 h), (3) mono-culture of *C. perfringens* in cell-free media from *P. distasonis* and *B. longum* subsp. *longum* after 24 h fermentation of LAC, BMO, and BMO + LAC. Late-log cultures were washed and adjusted to optical density 1.0 (OD_600nm_) using sterile saline (0.9% (*w*/*v*) NaCl) before inoculation. The co-culture inoculum consisted of OD-adjusted inoculum of *P. distasonis* and *B. longum* subsp. *longum*. Mono- and co-cultures contained 1% inoculum in pre-reduced media. Sterile water served as the negative control. Mono-culture experiments included two biological replicates, while co-cultures included three. After fermentation, samples were centrifuged at 7500 *g* for 1 min to produce supernatant and pellet samples. Sterile-filtered (using 0.22 μm syringe filters), cell-free media from mono-cultures of *P. distasonis* and *B. longum* sups. *longum* was re-inoculated with *C. perfringens* inoculum and allowed to propagate for 24 h. Sterile-filtered supernatant samples and pellets were frozen at −80 °C until further analysis.

### 3.5. Proton Nuclear Magnetic Resonance (^1^H NMR) Metabolomics

Supernatant samples were thawed for 30 min, vortexed, and filtered through prewashed 10k Millipore centrifugal filters (Amicon Ultra, Millipore Corp., Billerica, MA, USA) by centrifugation at 14,000× *g* for 30 min at 4 °C. A volume of 500 μL of filtered sample was transferred to an NMR tube with 60 μL of phosphate buffer (pH = 7.4) containing 0.23 mM DSS (3-(trimethylsilyl)-1-propanesulfonic acid-d6 sodium salt, Sigma-Aldrich, St. Louis, MO, USA) and 70 μL of D_2_O (deuterium oxide, 99.9%, Cambridge Isotope Laboratories, Andover, MA, USA). NMR spectra were acquired on a Bruker Avance 600 MHz NMR spectrometer (Bruker BioSpin, Gmbh, Rheinstetten, Germany) operating at a proton NMR frequency of 600.13 MHz and equipped with a 5 mm TXI probe. The 1D NOESY pulse experiment with pre-saturation of the spectral region containing the water peak (noesypr1d) was used with a recycle delay of 5 s. A total of 64 FIDs were acquired, and the acquisition parameters included 32 K complex data points, a spectral width of 7289 Hz (12.15 ppm), and an acquisition time of 2.25 s. Measurements were done at 298 K (25 °C).

### 3.6. Preprocessing of ^1^H NMR Data for Multivariate Analysis

An experimental window function with a line-broadening factor of 0.3 Hz was applied to all FIDs before Fourier transformation. The resulting spectra were manually phase corrected and automatically baseline corrected by polynomials using the Topspin 3.0 software (Bruker BioSpin, Gmbh, Rheinstetten, Germany). In Matlab (Version R2016a, The MathWorks Inc., Natic, MA, USA), data were referenced and scaled according to DSS at 0.0 ppm. Metabolites of interest were quantified using the Chenomx NMR suite software (Version 8.1 professional, Edmonton, AB, Canada) using the build-in library and an in-house library of 3′-SL and 6′-SL standards (Carbosynth, Berkshire, UK).

### 3.7. Carbon Mass in Co-Culture Experiment

Carbon mass was calculated for selected glycans and short-chain fatty acids using the following equation: g substrate × number of C-atoms × molecular mass of carbon/molecular weight of the substrate, as presented in [27]. The calculations were performed in Excel (Microsoft Office Professional Plus 2016).

### 3.8. DNA Extraction

Genomic DNA was extracted from bacterial pellets using the DNeasy Blood and Tissue kit (Qiagen, Valencia, CA, USA) according to manufacture protocol. Briefly, bacterial pellets were re-suspended in lysozyme solution and incubated at 37 °C for 30 min followed by Proteinase K treatment in AL buffer at 56 °C for 30 min. After addition of ethanol and vortex mixing the remaining steps were conducted according to the manufacturer’s instructions on DNA extraction from animal tissues. The extracted DNA was diluted 1:10 in RNAse free water and stored at −20 °C until further use.

### 3.9. Quantitative PCR

Quantitative PCR was performed using primers specific for the 16S rRNA gene of each of the three bacterial strains. The primers were previously described in [11]. Primers were ordered from Integrated DNA Technologies Inc. (San Diego, CA, USA). The bacterial DNA from the pure cultures were used as positive control and serial diluted in sterile Milli-Q water (10^−1^–10^−6^) to produce standard curves. The qPCR assay was run with a master mix containing forward (10 pmol) and reverse (20 pmol) primers, RNAse free water, and PowerUp SYBR Green Master Mix (Life Technologies, Carlsbad, CA, USA). A total of 3 µL of DNA or RNase free water for non-template controls (NTC) were added to each well to reach a final reaction volume of 20 µL. Two technical replicates were included per biological sample. The qPCR assay was run on a 7500 Fast Real-Time PCR System (Life Technologies, Dublin, Ireland), using the following program: 50 °C (2 min), at 95 °C (2 min), 40 cycles of: 95 °C (15 s), 56 °C (15 s), 72 °C (1 min). A melting curve analysis at 95 °C (15 s), 56 °C (1 min), 95 °C (30 s), and 56 °C (15 s) was performed to estimate the melting temperature of the amplicons. Melting curves with more than the expected single peak indicate unspecific PCR products and primer-dimer formation. The detection limit for inclusion of samples in the data analysis was determined to be 10^6^ copy numbers/mL culture. Delta log copy numbers per mL culture were calculated by subtracting the mean value of log copy numbers per mL in minimal media from each replicate value of log copy numbers per mL in carbohydrate media for each type of bacteria.

### 3.10. Data Analysis

Metabolite and glycan concentrations (mM) were presented as mean values with error bars representing standard error in bar charts. One-way ANOVA was applied to test the effect of bacteria on metabolite and glycan concentrations within each carbohydrate treatment.

In the co-culture study, growth data (delta log copy numbers per mL culture) were analyzed using two-way ANOVA to test the effect of carbohydrate treatment, bacteria (mono- or co-culture) as well as the interaction on growth. In the *C. perfringens* experiments, a two-way ANOVA was used to test the effect of carbohydrate and pretreatment of the media with B. *longum* subsp. *longum* as well as the interaction effect. All growth data are presented in barplots with lsmeans as bars and error bars representing standard error, as produced by an lsmeans fitted model. Tukey’s test for multiple comparisons was used to test the difference between treatment groups using a significance level of α = 0.05.

Before ANOVA analysis of metabolite or growth data, the datasets were checked for variance homogeneity and normality by visual inspection of Residual vs. Fitted and QQplots, respectively. Deviating samples were removed only if it could be argued they were indeed deviating substantially from e.g., other replicates of the same sample and if the sample contributed to non-normality of the dataset. Data analysis and visualization was performed in R (version 3.6.1, The R Foundation, Vienna, Austria).

Data is deposited as Appendix A.

## 4. Conclusions

In conclusion, the present study points toward a potential synergistic effect of lactose and BMO on metabolism of *B. longum* subsp. *longum*. Furthermore, it appears that commensalism between *B. longum* subsp. *longum* and *P. distasonis* in lactose-rich environment is beneficial for *B. longum* subsp. *longum* activity. Finally, metabolites from *B. longum* subsp. *longum* fermentation of BMO + LAC appeared to diminish *C. perfringens*’ ability to utilize BMO as a carbon source (potential postbiotic effects). Conducting additional research using other bifidobacteria strains capable of utilizing sialyated oligosaccharides for energy and metabolism is of major interest due to the possibility of enhancing the metabolic effect observed in the current study.

## Figures and Tables

**Figure 1 metabolites-10-00167-f001:**
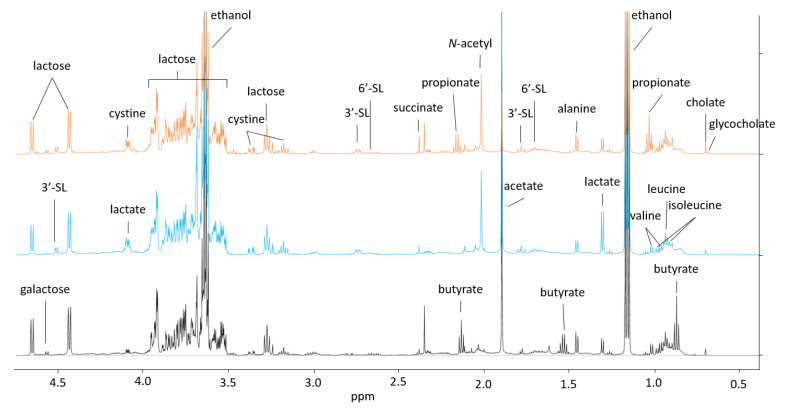
Aliphatic region of representative ^1^H NMR spectra obtained for supernatant from 24 h fermentation of minimal media containing 1% BMO + LAC. The colors of the spectra indicate the inoculated bacteria; orange (top): *Parabacteroides distasonis,* blue (middle): *Bifidobacterium longum* subsp. *longum* and black (bottom): *Clostridium perfringens*. BMO: bovine milk oligosaccharides, LAC: lactose.

**Figure 2 metabolites-10-00167-f002:**
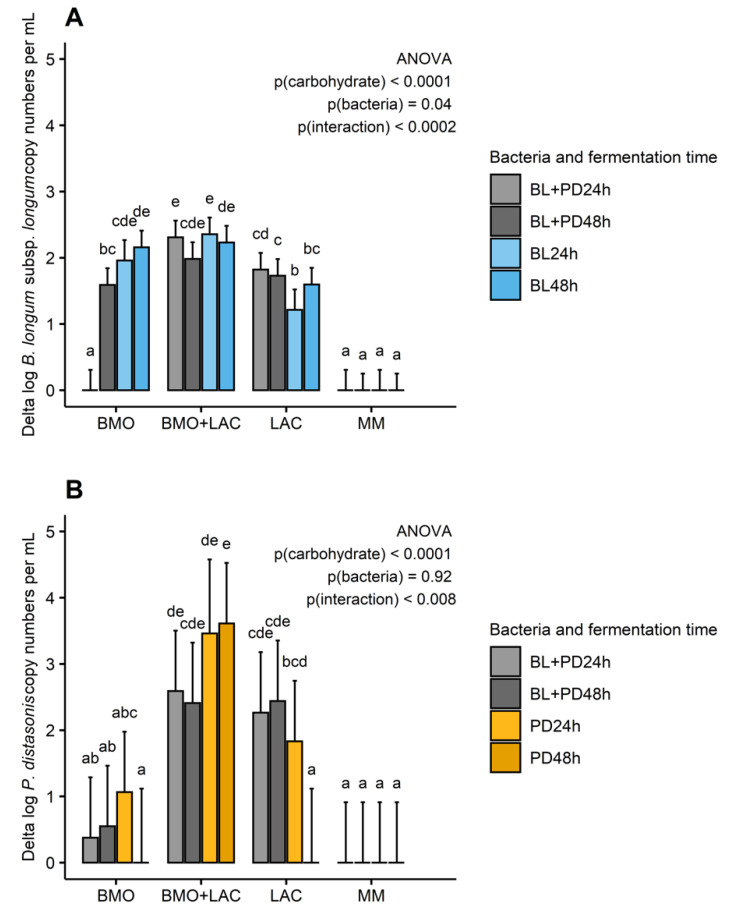
Delta growth of *B. longum* subsp. *longum* (**A**) and *P. distasonis* (**B**) in log10 copy numbers/mL culture compared to minimal media after 24 and 48 h fermentation of 1% BMO, BMO + LAC, or LAC media by *B. longum* subsp. *longum* (BL,-), *P. distasonis* (PD,-), and co-culture of *B. longum* subsp. *longum* and *P. distasonis* (BL + PD). Bars represent mean values and error bars represent standard error and are based on three biological replicates and two technical replicates. Significant effect of carbon source and bacteria on delta log copy numbers/mL culture was tested by ANOVA and Tukey’s HSD (honestly significant difference) was used for multiple comparisons between groups. *p* ≤ 0.05 indicates significant differences and different letters in each plot indicate significant differences. BMO: bovine milk oligosaccharides, LAC: lactose, MM: minimal media (no carbohydrate source).

**Figure 3 metabolites-10-00167-f003:**
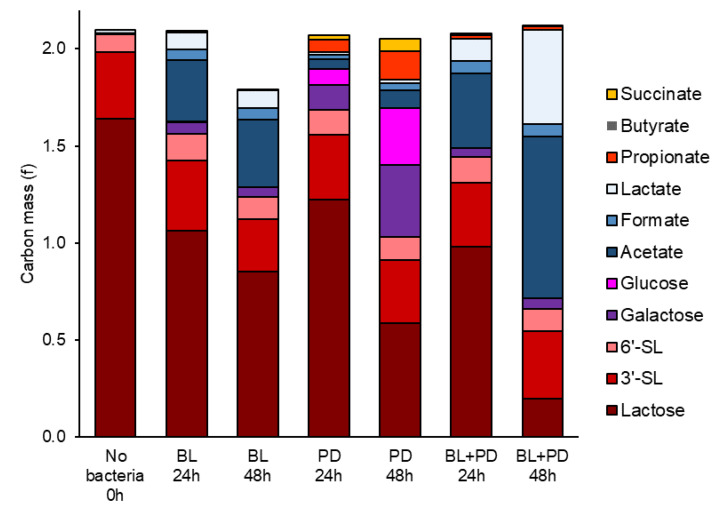
Carbon mass balance after 24 and 48 h fermentation of 1% BMO + LAC media by *B. longum* subsp. *longum* (BL), *P. distasonis* (PD) or co-culture of *B. longum* subsp. *longum* (BL, PD). Carbon mass was calculated as g substrate × number of C-atoms × molecular mass of C (12.01 u) divided by the molecular weight of substrate. Each stacked segment of the bars represents a mean value from three biological replicates. BMO: bovine milk oligosaccharides, LAC: lactose.

**Figure 4 metabolites-10-00167-f004:**
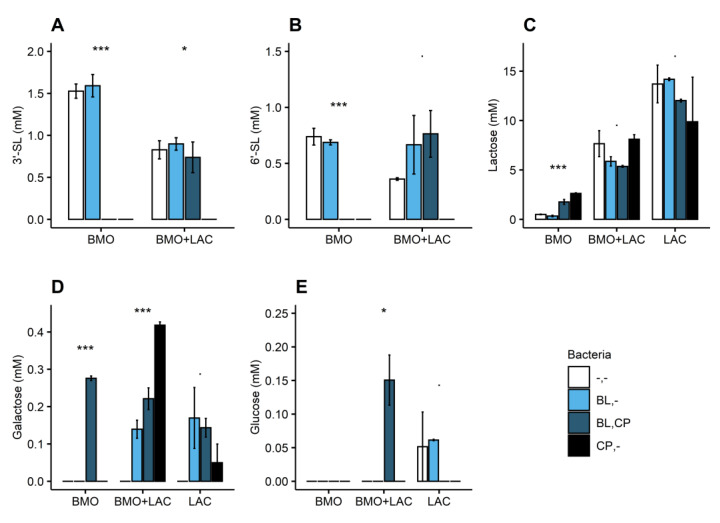
Glycan concentration (mM) in 1% BMO, BMO + LAC, and LAC media in uninoculated media (-,-) and after 24 h fermentation by *B. longum* subsp. *longum* (BL,-) or *C. perfringens* (CP,-) and *C. perfringens* fermentation of cell-free media from *B. longum* subsp. *longum* (BL,CP). (**A**) 3′-sialyllactose, (**B**) 6′-sialyllactose, (**C**) lactose, (**D**) galactose, (**E**) glucose. Bars represent mean values and error bars represent standard error and are based on two biological replicates. Stars indicate statistical significance by *p*-values from ANOVA testing on the bacteria effect within each carbohydrate treatment: *p* < 0.001: ***, *p* < 0.01: *, *p* < 0.1. BMO: bovine milk oligosaccharides, LAC: lactose.

**Figure 5 metabolites-10-00167-f005:**
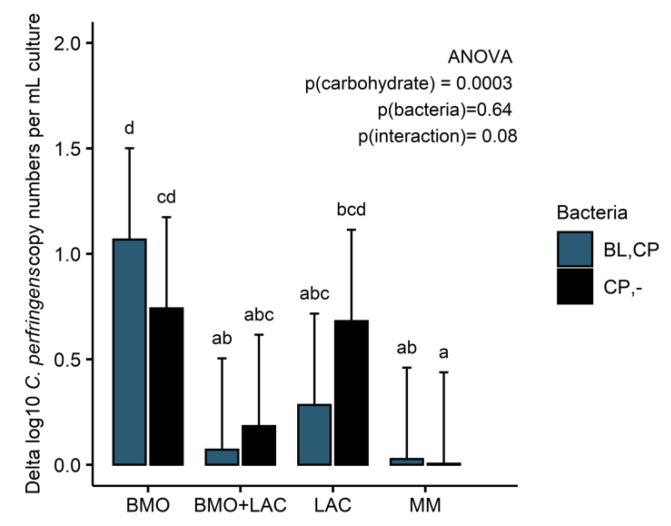
Delta *C. perfringens* growth in log10 copy numbers/mL culture compared to minimal media after 24 h fermentation of fresh 1% BMO, BMO + LAC, and LAC media (CP,-) and fermentation by *C. perfringens* in cell-free media obtained after 24 h fermentation by *B. longum* subsp. *longum* (BL,CP). Bars represent mean values and error bars standard error and are based on two biological replicates and two technical replicates. Significant effect of carbon source and bacteria on delta log copy numbers/mL culture was tested by ANOVA and Tukey’s HSD (honestly significant difference) was used for multiple comparisons between groups. *p* ≤ 0.05 indicates significant differences and different letters in each plot indicate significant differences. BL: *B. longum* subsp. *longum*, CP: *C. perfringens*, BMO: bovine milk oligosaccharides, LAC: lactose. MM: minimal media (no carbohydrate).

**Figure 6 metabolites-10-00167-f006:**
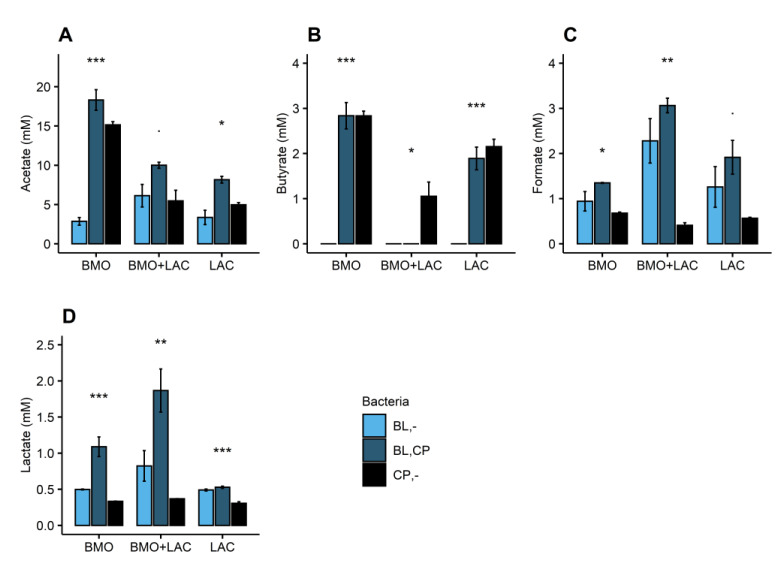
Metabolite concentration (mM) of (**A**) acetate, (**B**) butyrate, (**C**) formate, and (**D**) lactate after 24 h fermentation of 1% BMO, LAC, and BMO + LAC media by *B. longum* subsp. *longum* (BL,-), *C. perfringens* (CP,-) or *C. perfringens* fermentation in cell-free media obtained after 24 h fermentation by *B. longum* subsp. *longum* (BL,CP). Bars represent mean values and error bars represent standard error and are based on two biological replicates. Stars indicate statistical significance by *p*-values from ANOVA testing on the bacteria effect within each carbohydrate treatment: *p* < 0.001: ***, *p* < 0.01: **, *p* < 0.05: *, *p* < 0.1. BL: *B. longum* subsp. *longum*, CP: *C. perfringens*, BMO: bovine milk oligosaccharides, LAC: lactose.

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
