# Peer review of "Metabolic Effects of Bovine Milk Oligosaccharides on Selected Commensals of the Infant Microbiome—Commensalism and Postbiotic Effects"

_metabolites, 2020, doi:10.3390/metabo10040167_

Round 1

Reviewer 1 Report

The Manuscript of Jakobsen et al presents information about the metabolism of 2 infant gut bacterial strains on bovine milk oligosaccharides +- lactose. Moreover, it shows effects of culture supernatant of these commensals grown on these carbohydrates on toxin-producing C. perfringens.

If the results are interesting for the scientific community, the relevance for the society is not so clear. The motivation of this work should be better stated at the beginning of the introduction, as well as the concrete aims at the end of the introduction (to describe an interaction is a bit vague as an aim). Moreover, the authors stated in the introduction that LAC inhibits C. perfringens growth in coculture in a previous work, but in the given reference (fig1 and 2 of ref. 11) it does not. Moreover, I do not think “postbiotic” is really an appropriate term, as it would imply a benefice for a host, which is not to represented in in vitro cultures. Less toxin-producing bacteria could be a real gain for the host, but is not necessarily

The English is correct and the formulation of the sentences is mainly clear, however I am unsure if the two aims of the authors should not be presented in 2 different articles. As it is, it represents a lot of data (presented in supplemental material as tables, not easily comparable to the graphic representation in fig1 or between each other, and partly redundant), but still does not really answer the “burning” questions of the mechanisms behind (no real “deciphering”). I would therefore recommend the authors to perform some complementary experiments/measurements and then submit 2 separate articles:

  • one concentrating on the metabolism of BMO+-LAC by commensals (in single and coculture), going a bit further as done here. For example, authors mention that B. longum is unable to use the main compounds of BMO (I then wonder what are the 57% w/w rest of BMO made of if other oligosaccharides represent only traces?). It would then be really interesting to further analyse which components of BMO are used by B. longum, as this one is growing quite well on it. The authors give some hypotheses in the discussion, they could check for. Another point is stated by the authors themselves: they say that more work is needed to understand why in presence of both C-sources (BMO and LAC) this bacterium is growing better and its metabolism is more active. They mention a possible “positive feedback” on LAC utilization in presence of BMO: is something similar already known, in other species? Here would a reference be helpful. And this definitively would be an exciting question to answer. Moreover, the authors report an interesting increase in metabolism when both strains are cocultured on LAC or even more on LAC+BMO. They discuss that BL would not use the monomers released by PD in presence of LAC. However, in Fig1 it is explicit that the monomers produced by PD are used further in the coculture (low quantities of them). Could the authors discuss on this and with it explain why, more fermentation products are then produced. How does this “commensalism” work? It would be really interesting to characterize the mechanisms.
  • One concentrating on the inhibition of the pathogen C. perfringens. Here I wonder why the authors tested both PD and BL separately and not as coculture to condition the medium for CP, as they showed a stronger activity of both strains when cocultured. In their previous article, the authors showed an inhibition of CP by BMO in coculture. Here they show that in single culture, as well as cultured on BL (or PD)-conditioned-medium, CP is able to use BMO (both 3’ and 6’ SL) efficiently. Only in the presence of LAC and on BL-conditioned medium is CP’s use of the SLs inhibited. The author discuss if this effect could be due to the lower pH or organic acids of BL-conditioned medium (on BMO +LAC) as the sialidases are pH-sensitive and their genes could be regulated by the virulence gene expression. I would recommend the authors to perform complementary experiments, quantifying the gene expression of both sialidases and Vir genes in culture with adjusted pH or addition of organic acids compared to BL-conditioned media to decipher the mechanisms behind the observed inhibition. Also it would be particulraly interesting to know if the toxin production of CP is influenced by the different carbohydrate sources and cocultures (if so a postbiotic effect could be indeed mentioned).

Moreover, authors need to check all legends for needed abbreviations (MM for example), number of replicates (and which kind: biological or technical?). Also, in the SM so formulation are false: no” bars” in the tables! If giving the values as mean, it is necessary to give the standard deviation (or another indication of variance), then only would SM Table 3 make sense complementing Fig 1 in which SD can for sure not be represented. Pay attention if mentioning a SM, first should come SM Table 1, then SM Table 2. In SM Table4 growth is mentioned: after how long? Also SM Table  5 and 6 are not referred to.

Another point that should be improved concern the statistics: in §3.10 statistics used are not clearly stated. First it should be mentioned how the data was tested for distribution. ANOVA is known to have restricted conditions of use. The author mentioned using ANOVA in SM Tables, but in the first 3 tables there are no statistic results: be more precise: only SM Table 5&6! What about the figures? Particularly Fig4, with 2 factors (bacteria and carbohydrate): what has been exactly tested here, how? Fig3 and 6 have no indication of statistics: why?

Finally, here a list of some minor things to check:

Title of §2.1 is badly formulated and does not represent the results shown in this § (namely the single culture of both commensal strains on the 3 different C-sources combinations).

  • 2.2: title not representing the results. OK for the metabolism part, not really for the growth of B. longum (only significant after 24h and only on LAC). Remove L 139-140 as this is either false (no coculture in SM Table5) or redundant with L127. L142; fig 1 and SM 2&3, otherwise no comparison possible. L149-150 belongs to §2.1 (single culture).

L241: …incubation at 37°C in a heating…

L249: 12 &24h or 24 and 48h? all data shown are at 24 or 48h…

L286: why is “ref 26” bigger than the rest of the text?

L294: diluted with what?

L306: remove “was”

L307-308: as far as I know a melting curve is done by performing a gradient in temperature from high to low by decreasing temperature of X °C per Y seconds. Please improve.

Fig1: I am missing a bar of time point 0, as a visual control of which sources of C are present at the start of the cultures. Also, which other kinds of C-sources and at which (total) concentration are present in the media?

Fig2: try and use a different intensity of color for the 2 time points (at least in the pdf version I got, there was no seeable difference)

Reviewer 2 Report

The manuscript looks at some "potential" interactions between commensal and postbiotic microbes in the infant gut by in vitro co-cultures. the experimental design is sound and the conclusions mostly support the authors' hypotheses. Please address the following comments and resubmit. 

  • Increase the size of the legends in all your figures (especially Figure 1) so that the color codes are legible.
  • Your claim of the inability of B. longum to metabolite galactose and glucose monomers is not strong. While closely related species can selectively uptake certain metabolites instead of these, a claim that your organism of interest is NOT capable of utilizing these sugars needs much stronger experimental verification. Also see articles (https://aem.asm.org/content/74/22/6941/article-info and https://www.ncbi.nlm.nih.gov/pubmed/21630463) where Bifidobacterium longum is seen to utilize these sugars. 

Reviewer 3 Report

The present study has an important and interesting topic: the authors examined the interaction between Bifidobacterium longum subsp. longum and the commensal Parabacteroides distasonis, by culturing them in mono- and coculture with bovine milk oligosaccharides. The major conclusions were supported by the experimental results. There are a few issues the authors need to address to improve the manuscript.

 (1) The authors should show the representative 1H NMR spectra of bacteria in the study. The sensitivity of NMR is not good enough for metabolomics analysis of bacterial strains. Especially, the authors used simple extraction protocol without homogenization or methanol extraction solvent.

(2) What is the purpose for qPCR? The authors should do real time PCR and run the gel or 16S rRNA gene sequencing to confirm the bacteria. In addition, the authors should show the related data in the supplement.

(3) Figure 1: Why the authors use the carbon mass as unit instead of the amount of the metabolites? The colors of glucose and galactose are similar and confusing.

(4) Figure 2: The authors should consider the growth curve that will be clearer.

(5) Some supplementary Tables that contain meaningful data should be placed in the regular body text.

Round 2

Reviewer 1 Report

Thank you for your answers and your modifications to your article, which is now clearer and better. I have no further suggestions.

Reviewer 3 Report

The manuscript has been significantly improved and now warrants publication in Metabolites.